# Associations between Isolation Source, Clonal Composition, and Antibiotic Resistance Genes in *Escherichia coli* Collected in Washington State, USA

**DOI:** 10.3390/antibiotics13010103

**Published:** 2024-01-20

**Authors:** Mary Jewell, Erica R. Fuhrmeister, Marilyn C. Roberts, Scott J. Weissman, Peter M. Rabinowitz, Stephen E. Hawes

**Affiliations:** 1Department of Epidemiology, School of Public Health, University of Washington, Seattle, WA 98195, USA; mj49@uw.edu (M.J.); hawes@uw.edu (S.E.H.); 2Department of Environmental and Occupational Health, School of Public Health, University of Washington, 3980 15th Ave NE, Seattle, WA 98195, USA; efuhrm@uw.edu (E.R.F.); peterr7@uw.edu (P.M.R.); 3Department of Civil and Environmental Engineering, University of Washington, 3760 E. Stevens Way NE, Seattle, WA 98195, USA; 4Division of Infectious Disease, Seattle Children’s Hospital, Seattle, WA 98105, USA; scott.weissman@seattlechildrens.org; 5Center for One Health Research, University of Washington, 3980 15th Ave NE, Seattle, WA 98195, USA

**Keywords:** *Escherichia coli*, antibiotic resistance, whole genome sequencing, antibiotic resistance genes

## Abstract

Antimicrobial resistance (AMR) is a global health problem stemming from the use of antibiotics in humans, animals, and the environment. This study used whole-genome sequencing (WGS) of *E. coli* to explore patterns of AMR across sectors in Washington State, USA (WA). The WGS data from 1449 *E. coli* isolates were evaluated for isolation source (humans, animals, food, or the environment) and the presence of antibiotic resistance genes (ARGs). We performed sequence typing using PubMLST and used ResFinder to identify ARGs. We categorized isolates as being pan-susceptible, resistant, or multidrug-resistant (MDR), defined as carrying resistance genes for at least three or more antimicrobial drug classes. In total, 60% of isolates were pan-susceptible, while 18% were resistant, and 22% exhibited MDR. The proportion of resistant isolates varied significantly according to the source of the isolates (*p* < 0.001). The greatest resistance was detected in isolates from humans and then animals, while environmental isolates showed the least resistance. This study demonstrates the feasibility of comparing AMR across various sectors in Washington using WGS and a One Health approach. Such analysis can complement other efforts for AMR surveillance and potentially lead to targeted interventions and monitoring activities to reduce the overall burden of AMR.

## 1. Introduction

The growing prevalence of antimicrobial resistance (AMR) represents a significant global health problem, undermining progress in healthcare and agricultural food production around the world. The CDC estimates that more than 2.8 million antibiotic-resistant infections occur each year in the United States, resulting in 35,000 deaths [1]. AMR infections pose a threat to people all over the world, and they make crucial healthcare procedures, like joint replacements and organ transplants, potentially life-threatening [1].

AMR is caused by the use and overuse of antibiotics in humans and animals, as well as the contamination of the environment antibiotic residues, AMR genes, and bacteria. Previous studies indicate that the spread and proliferation of resistant bacteria is driven by interactions between these sectors [2,3,4,5,6]. Due to the interdependent nature of AMR in humans, animals, and the environment, it is advisable to use a One Health approach to address the problem [6]. One Health is based on the interconnected nature of humans, animals, and the environment, acknowledging that we share many of the same health exposures and infectious diseases [6]. A One Health approach to AMR involves the collaborative efforts of many health professionals across disciplines to prevent the spread of antimicrobial-resistant pathogens [6].

*Escherichia coli* (*E. coli*) is considered by the World Health Organization to be one of the top organisms of international concern related to AMR [7,8]. *E. coli* is an opportunistic pathogen, which is widespread among animals and the environment, and it is characterized by its genetic diversity due to its flexible gene pool of mobile genetic elements and genomic islands [6,9,10]. There have been many studies examining resistance in *E. coli* in specific settings, such as human clinical samples [11,12,13], livestock and agriculture [14,15], and pre-packaged food items [4,16]. However, fewer studies examine AMR from a One Health perspective, comparing resistance across multiple sources. Studies that do investigate this have shown that meat, particularly poultry, may be a source of *E. coli* infections in humans [17,18,19]. In Washington State, research has separately reported the presence of antimicrobial-resistant *E. coli* in the Salish Sea ecosystem [5] and in outpatient clinical samples isolated from Washington State Public Health Laboratories [11,20].

The goal of this study was to use publicly available *E. coli* genome sequences to evaluate the relationship between sample source and the presence of ARGs in *E. coli* isolates from Washington State. To explore this question, we used multi-locus sequence typing (MLST) to assign genomes to known sequence types (STs). We then measured the degree of association between the isolation source and the relative distributions of STs and noted the frequency of ARGs present among all *E. coli* samples from Washington State, and among samples of the same sequence type (ST). This study involved conducting a bioinformatic analysis of publicly available genomic data, but it did not contribute any new *E. coli* strains.

## 2. Results

### 2.1. Characteristics of E. coli Isolates

In total, the NCBI GenBank database provided genomic data for 1517 *E. coli* isolates from Washington State, USA. Of these, 68 were eliminated due to poor read quality or missing information about bacterial isolation source. Of the remaining 1449 isolates, 296 (20.4%) were isolated from animals, 327 (22.6%) were isolated from environmental sources, 54 (3.7%) were isolated from food, and 772 (53.3%) were isolated from humans. Examples of common specific isolation sources are listed in Table 1. Isolates were collected between 1947 and 2022. There were 109 isolates (7.5%) collected before 2005, 343 isolates (23.7%) collected between 2005 and 2015, and 579 isolates (40.0%) collected after 2015. There were 418 isolates (28.8%) with no data for collection year (Table 1). All isolates were collected within Washington State, but most isolates (70.5%) were missing additional geographic information. Of those with any additional geographic information, the only two Washington counties represented were King County (n = 264, 18.2%) and Thurston County (n = 164, 11.3%).

### 2.2. Sequence Typing of E. coli Isolates

The 1449 isolates were distributed among 341 different Sequence Types (STs). The STs with the most isolates overall were ST131 (n = 131, 9.0%), ST73 (n = 80, 5.5%), and ST95 (n = 65, 4.5%). Several of the most common STs in human isolates are known to be lineages of extraintestinal pathogenic *E. coli* (ExPEC), including STs 131, 69, 95, and 73, which are associated with common urinary tract and bloodstream infections (34). ExPEC group STs were also commonly found in animal and environmental isolates, including STs 10, 117, 69, 73, and 58. Most STs (n = 172, 50.5%) were represented by only one isolate, while 59 STs (17.3%) were found in only two isolates. There were 66 isolates (4.6%) whose ST could not be identified (“-”). The isolates with unidentified STs came from all four isolation sources, but most commonly (59%) from environmental sources. Human isolates had the lowest proportion of samples with an unidentified ST.

The most common STs differed between isolation sources (Figure 1). Isolates from animal and food sources had three of their six most frequent STs in common: ST21, ST11, and unidentified ST. STs that were common in animal, food, and human sources included STs 11 and 372, and STs that were common in both animal and human sources included STs 10 and 127. Animals and humans share their most common ST, ST21, while in humans, the most common was ST131, and in environmental isolates, an unidentified ST was the most common. The most common STs among environmental isolates were least likely to be shared with other isolation sources; of the 12 most common STs in environmental samples, nine were unique to environmental samples.

We compared the distribution of isolates in the six most common STs overall to assess whether the relative frequency of isolates in each of these STs was similar in all of the four isolation sources. We compared the relative frequency of ST131, ST73, unidentified ST, ST95, ST12, and ST69 in animal, environmental, food, and human sources. We found that there was evidence of a difference in the relative frequency of STs between the four isolation source categories (*p* < 0.001).

### 2.3. Predicted Antimicrobial Resistance

In total, 60% of isolates were predicted to be pan-susceptible, while 18% were predicted to be resistant, and 22% were predicted to be multidrug-resistant. The resistance pattern varied between the four isolation sources (*p* < 0.001). Most resistance genes were detected in isolates from humans, with 54% of these samples categorized as either resistant or multidrug-resistant, followed by animals, with 37% of samples either resistant or multidrug-resistant (Figure 2). About one third (33%) of human isolates were predicted to be multidrug-resistant, the most of any other isolation source. Environmental isolates showed the least antimicrobial resistance, with only 12% of environmental samples categorized as either resistant or multidrug-resistant.

Within each of the most common STs, the resistance pattern also varied between the four isolation sources (Figure 2). The difference in the proportion of isolates predicted to be pan-susceptible, resistant, or MDR between isolation sources was statistically significant among isolates of ST131 (*p* = 0.003) and ST73 (*p* = 0.002). Within the other most common STs (ST95 and unidentified ST), there was not significant evidence of association between antibiotic resistance and isolation source within each ST.

### 2.4. Antimicrobial Resistance over Time

There were three peaks in isolate collection year over time: in 1985, 2011, and 2019. Based on these peaks, we broadly grouped isolates into time periods defined as 1947–2004, 2005–2015, and 2016–2022. Comparing isolates across these three time periods, our data show that the greatest levels of resistance overall were seen in the period 2005–2015. However, this relationship is different within each isolation source (Figure 3). In humans, the greatest proportion of resistant and multidrug-resistant isolates appear in 2005–2015, but in animal isolates, the next largest isolation source, the greatest proportion of resistant and multidrug-resistant isolates were collected in 1947–2004. In environmental and food isolates, small sample sizes make it difficult to discern a clear pattern.

A multinomial logistic regression shows that, as collection year increases, the odds of an isolate being predicted to be resistant vs. pan-susceptible decline by about 2% while holding isolation source constant (*p* < 0.001). However, the odds of an isolate being predicted to be multidrug-resistant vs pan-susceptible increase about 1% every year while holding isolation source constant (*p* < 0.001). Isolates from animals have 58% lower odds of being categorized as multidrug-resistant compared to isolates from humans within the same collection year (*p* < 0.001). Similarly, compared to human isolates, environmental samples have 85% lower odds of being categorized as multidrug-resistant (*p* < 0.001) and food samples have 87% lower odds of being categorized as multidrug-resistant (*p* < 0.001).

### 2.5. Antimicrobial Resistance Profiles

There were 23 unique resistance profiles based on binary resistance indicators for five key antimicrobial agents: ampicillin, ceftazidime, ciprofloxacin, gentamicin, and tetracycline. The most common resistance profile overall was a lack of resistance genes (pan-susceptible) for all five antimicrobial agents (64% of isolates), followed by resistance genes for ampicillin alone (6% of isolates) and tetracycline alone (5% of isolates) (Appendix A). Around 1% of isolates had antimicrobial resistance genes (ARGs) for all five profile drugs. Human isolates had the greatest number of unique resistance profiles, followed by environmental isolates. When comparing the distribution of isolates in the six most common profiles overall (pan-susceptible to all five drugs, resistance genes for ampicillin alone, resistance genes for tetracycline alone, resistance genes for ampicillin and ciprofloxacin, resistance genes for ampicillin and tetracycline, and resistance genes for ciprofloxacin alone), there was evidence of a significant difference in the frequency of resistance profiles between the four isolation source categories (*p* < 0.001) (Figure 4).

### 2.6. Distribution of Specific Antimicrobial Resistance Genes

The prevalence of ARGs varied considerably between the four isolation sources, with some genes being common between all four sources and several found only in isolates from a single source (Appendix A). Fourteen different resistance genes were present in all four isolation sources. The genes most frequently shared in all four sources conferred resistance to sulfonamides (*sul2*, n = 243), aminoglycosides (*aph(3)-Ib*, n = 233; *aph(6)-Id*, n = 232; *aadA5*, n = 206), and tetracyclines (*tet*(B), n = 177). The most common genes conferring resistance to beta-lactamases were *bla*_TEM-1B_ (n = 365), which was present in isolates from animals, humans, and the environment; *bla*_CTX-M-15_ (n = 95), which was present in isolates from humans and the environment; and *bla*_OXA-1_ (n = 88), which was only present in isolates from humans.

## 3. Discussion

This study of 1449 *E. coli* isolates from Washington State found a total of 341 unique STs, and the most common STs in isolates from animal, environmental, food, and human sources differed, with some STs being found more commonly or exclusively in one isolation source. We also found a significant difference in the proportion of isolates that were predicted to be resistant and multidrug-resistant when comparing genes present in samples from the four isolation sources. Samples from humans showed the most resistance genes, followed by animals, food, and lastly environmental samples.

The distribution of STs within human isolates is unsurprising, with many of the most common STs being known to be ExPEC lineages, which are associated with human infections. Food can be a vector of transmission of ExPEC, but interestingly, in our data, none of the most common food STs were ExPEC-group STs. However, several ExPEC lineages, including STs 10, 58, and 117, were commonly found in environmental and animal isolates, potentially suggesting other means of transmission to humans. One study suggests that ST372, one of the most prevalent lineages in dogs, is a potential zoonotic pathogen that causes extraintestinal infections in humans [21], and our data support this, as ST372 was among the most common STs in both humans and animals.

Even though environmental isolates were not the largest group, they had the greatest number of unique STs and the greatest number of isolates with an unidentified ST. This suggests that environmental isolates are highly diverse and may include isolates not normally associated with the other three sources. The number of ExPEC lineages found in environmental isolates also indicates that they represent a potential reservoir of disease-causing bacteria, even though resistance gene frequency overall was lower in environmental isolates compared to humans and animals. Environmental *E. coli* samples are not as thoroughly studied as human clinical samples, but there have been some studies examining *E. coli* in marine environments [5,22]. These studies support the notion that environmental sources are an overlooked reservoir of epidemiologically relevant bacteria. Unfortunately, the small sample of environmental isolates in this database limited the analysis of factors contributing to AMR in the environment. Further studies focusing more on the impact of antibiotic use in agriculture and aquaculture would strengthen this field by expanding understanding of AMR beyond animal and human studies.

Our overall AMR results support the hypothesis that there is a relationship between isolation source and the presence of ARGs in *E. coli* samples from Washington State. The proportion of isolates predicted to be resistant and multidrug-resistant was highest in human samples, which is consistent with other research that shows high levels of antimicrobial resistance in human isolates [11,23]. Past studies have shown that demographic factors such as patient age and sex, as well as clinical factors like hospitalization status, are also associated with high levels of drug resistance seen in bacterial isolates from humans [11,23]. However, while most human isolates in our data are from clinical samples, additional metadata about the reason for sampling, patient demographic factors, and geocodes are not available, illustrating one of the limitations of this data source for epidemiologic studies. There are several reasons why sampling bias could account for much of these results. For example, it would be expected that isolates from both humans and animals would come from diseased hosts and some of the isolates represent the causative pathogen for a disease case, which is a different selection process than how environmental and perhaps food samples were collected.

Our study found that genes conferring antibiotic resistance to broad-spectrum antibiotics such as beta-lactams, aminoglycosides, and tetracyclines are present in all four isolation sources. However, as Ludden et al. [10] point out, while this supports the ubiquity of these genes, it does not necessarily provide evidence of a recent transfer between isolation sources. Their research found distinct mobile genetic elements between humans and livestock, suggesting limited genetic transfer between isolation sources [10]. Similarly, our data showed that the distributions of resistance profiles in each of the four isolation sources were distinct from one another, potentially indicating separate silos of antibiotic resistance. On the other hand, Jakobsen et al. [17] suggest that, in their data, human antibiotic consumption alone does not account for the resistance patterns seen in human isolates. Studies also suggest that food animals and meat may be an important source of antibiotic resistance in humans. Though our study is not able to determine how drug resistance spreads between isolation sources, it does show that antimicrobial resistance is a problem that requires a One Health approach.

Our study also showed that the proportion of isolates that are predicted to be resistant or multidrug-resistant compared to pan-susceptible has changed over time since 1947, as would be expected, since antibiotic use has increased since the 1950s. Overall, the likelihood of an isolate being multidrug-resistant increases every year, but the likelihood of an isolate being resistant decreases slightly every year. Isolates from humans were more likely to be predicted to be resistant or multidrug-resistant over time compared to animal, food, and environmental isolates. Other studies also provide some evidence that resistance changes over time. For example, one meta-analysis of 15 studies shows an increase in resistance to ciprofloxacin, one of the most widely used broad-spectrum antibiotics, between the years 2000 and 2018 [24]. While that analysis did consider isolates from humans, animals, food, and the environment, it did not differentiate between isolation source when examining rates of resistance over time. That analysis also included data from all over the world, and it is known that antibiotic consumption is significantly different in low- and middle-income countries (LMICs) compared to high-income countries. Our findings suggest that there may also be a significant trend in Washington, but it is important to differentiate between bacterial isolation sources when considering trends over time.

This study has several limitations. Due to the non-systematic nature of how sequences are deposited in the NCBI GenBank database, these data cannot be interpreted as defining the prevalence of resistance in any source type in Washington. As a whole, these data were largely from human sources. Small sample sizes in some other isolation sources, such as food, limited some comparisons. Categorizing samples by isolation source also simplified possible differences in microbial ecology between different ecological niches, potentially obscuring differences between and within groups. For example, there were a small number of samples from wild animal species, likely from zoos, which we categorized as animals along with livestock and domesticated animals. However, this heterogeneity masks potential relationships that may exist between specific isolation sources; for example, the microbiome of retail meat may have more in common with livestock than with wild animal species.

The available metadata also reflect the limitations of using pathogen genomic data in epidemiological studies. Although the integration of these fields for pathogen surveillance is of rising interest, some epidemiologically relevant data fields, such as the reason for collecting a sample, are scarce or missing in genomic metadata. In our study, there were also many samples that were missing relevant metadata about the timing and location of sample collection. Lastly, there are inherent limitations with the use of genotypic data without phenotypic confirmation; we do not know which resistance genes present in our sample are functional, and genotypic prediction relies on the completeness of the reference database. Although most studies indicate that the presence of ARGs closely corresponds with phenotypic resistance [25,26,27], some more recent studies have found a lack of correlation between the results of phenotypic susceptibility testing and the ARGs present according to whole genome sequencing [5]. It is possible that some of the resistance genes identified in this study do not confer functional resistance, and some functional resistance genes may not have been identified. However, ResFinder is a large, frequently updated database, and several benchmarking studies show that it has a high accuracy and performance [28,29], particularly for *E. coli* genomes [30].

## 4. Conclusions

Our study found that the proportion of isolates predicted to be resistant varied significantly by isolation source, with isolates from humans showing the greatest resistance, and isolates from environmental sources showing the least resistance. Our results build on existing evidence that the problem of antibiotic resistance is interdisciplinary, requiring surveillance, prevention, and mitigation efforts across multiple sectors. Our study highlights the need to further characterize bacterial isolates from a One Health perspective, both in Washington State and elsewhere. As the speed of sequencing technologies and the number of available isolates continue to increase rapidly, research into genomic surveillance methods will become increasingly important.

## 5. Materials and Methods

### 5.1. Data Collection

Enterobase [31] is a large public database of submitter-supplied data, which collates all the complete genomes and Illumina paired-end reads of enteric bacteria from the NCBI Sequence Read Archive, together with available metadata. We searched Enterobase for all genomes of the species *E. coli* within the state of Washington, USA, and categorized them according to their metadata as being isolated from either animal, environmental, food, or human sources. Samples with no information about their host or isolation source were excluded. We downloaded raw sequence files from NCBI in FASTQ format and used Trimmomatic [32] to remove Illumina adapters and clean the sequence files. We used options for a sliding window averaging across four bases with an average quality of 20, and we kept sequences with a minimum length of 25 bases. NCBI Accession numbers of the isolates included in this study are available in File S1.

### 5.2. Sequence Typing

We used Megahit [33] for *de novo* assembly of raw reads prior to sequence typing. We performed sequence typing using PubMLST [34] with the Achtman MLST scheme [35] to identify the sequence type (ST) of each bacterial isolate based on differences in DNA sequences of seven housekeeping genes. Isolates were labeled with an unidentified ST (“-”) if they had a novel allele, a novel allele combination, or one or more alleles with only a partial match.

### 5.3. Identification of AMR Genes

ResFinder [36] is a bioinformatics tool that uses BLAST to identify antimicrobial resistance genes in WGS data using a manually curated database of antimicrobial resistance genes identified in WGS data. We used ResFinder to identify AMR genes in all isolates, using raw reads to improve sensitivity and options for both acquired resistance genes and chromosomal mutations. The threshold for gene detection was set to 90% identity and 100% coverage.

### 5.4. Definitions of Drug Resistance

We categorized isolates as being pan-susceptible, resistant, or multidrug-resistant (MDR) based on guidelines adapted from the European Centre for Disease Prevention and Control (ECDC) and the Centers for Disease Control and Prevention (CDC) [36]. These guidelines define 15 epidemiologically significant categories of antimicrobial agents, as follows: aminoglycosides, antipseudomonal penicillins + beta-lactamase inhibitors, carbapenems, extended-spectrum cephalosporins (3rd and 4th generation cephalosporins), cephamycins, fluoroquinolones, folate pathway inhibitors, glycylcyclines, monobactams, penicillins, penicillins + beta-lactamase inhibitors, phenicols, phosphonic acids, polymyxins, and tetracyclines. Pan-susceptible isolates are defined as isolates that do not carry resistance genes for any antimicrobial agents in any of the categories. Resistant isolates are those that carry resistance genes to at least one antimicrobial agent in up to two different categories. Multidrug resistance is defined as carrying resistance genes for at least one agent in three or more antimicrobial categories.

### 5.5. Resistance Profile

We developed an antimicrobial resistance profile for a genome based on binary presence of resistance gene(s) to five key antimicrobial agents representative of their class: ampicillin, ceftazidime, ciprofloxacin, gentamicin, and tetracycline. These antibiotics were selected because they are commonly used in human and animal medicine. Predicted resistance to each of these drugs (the presence of resistance genes to that antibiotic) is represented by a 1, while predicted susceptibility is represented by a 0, resulting in a five-digit resistance profile. We compared the distribution of resistance profiles in each of the four isolation sources.

### 5.6. Statistical Analysis

We assessed differences in the number of isolates in each ST between isolation sources using Fisher’s exact test. We used a chi-square test to compare the numbers of pan-susceptible, resistant, and multidrug-resistant isolates in each isolation source. To examine changes in levels of resistance over time, we conducted multinomial logistic regression to analyze the relationship between antimicrobial resistance category (either resistant or multidrug resistant compared to pan-susceptible) and collection year. We included isolation source in the model with isolates from humans as the reference category. All effects were evaluated using a significance level of 0.05.

### 5.7. Ethical Approval

This research was determined to not involve human subjects as defined by the Institutional Review Board (IRB) of the University of Washington, and therefore did not require IRB approval or exemption.

## Figures and Tables

**Figure 1 antibiotics-13-00103-f001:**
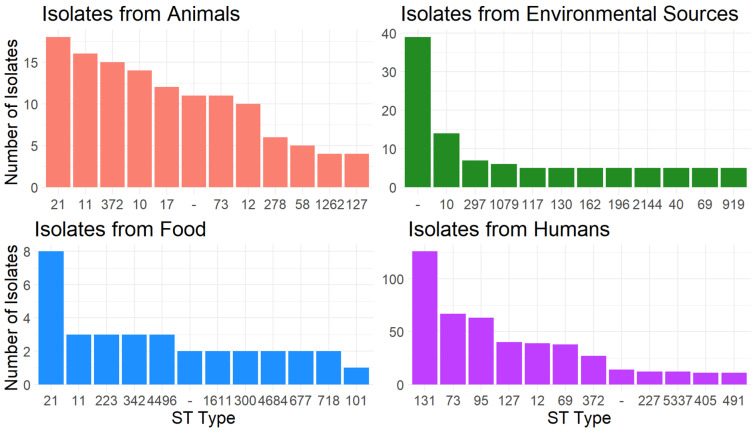
The number of isolates in each of the 12 most common STs within each isolation source. Common STs found in two or more isolation sources include ST10, ST11, ST12, ST21, ST69, ST73, ST127, ST372, and unidentified ST (-).

**Figure 2 antibiotics-13-00103-f002:**
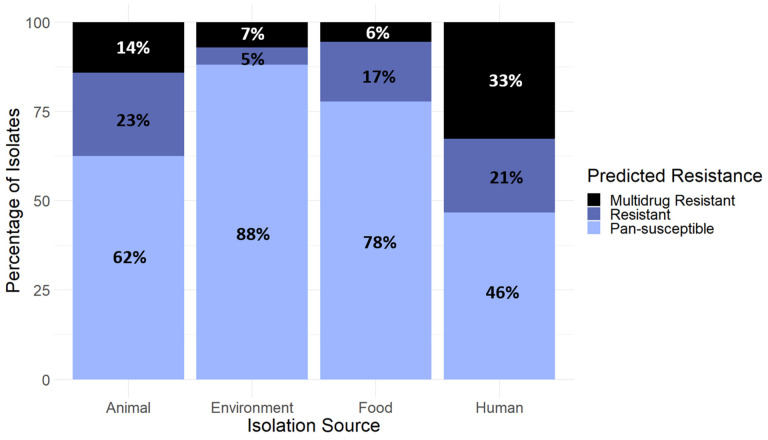
The proportion of isolates within each isolation source that are categorized as pan-susceptible, resistant, or multidrug-resistant. Due to rounding, not all percentages add to 100%.

**Figure 3 antibiotics-13-00103-f003:**
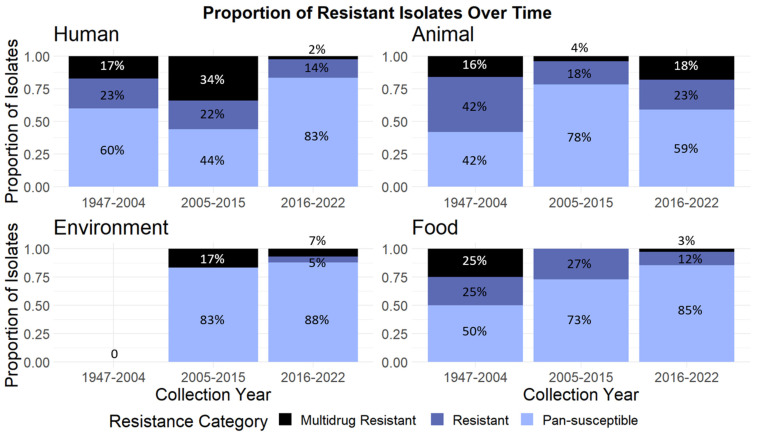
The proportion of isolates collected in each time period (1947–2004, 2005–2015, or 2016–2022) that are categorized as pan-susceptible, resistant, or multidrug-resistant. Due to rounding, not all percentages add to 100%.

**Figure 4 antibiotics-13-00103-f004:**
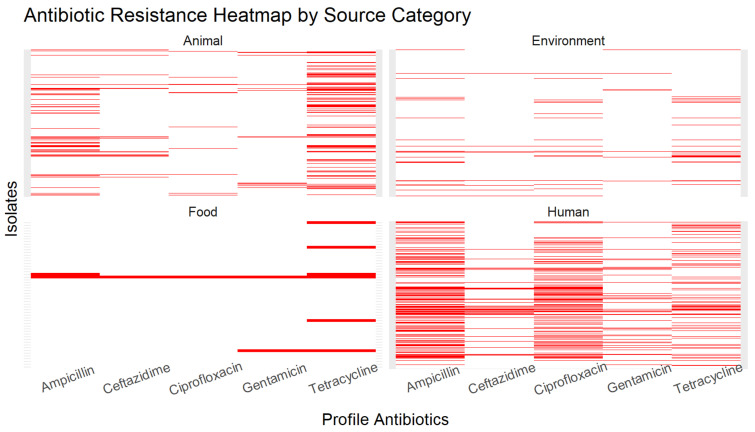
Heatmap representing the antibiotic resistance profiles of *E. coli* isolates. Rows represent individual isolates and columns represent each antibiotic included in the resistance profile. Red blocks indicate predicted resistance, while white blocks indicate predicted susceptibility.

**Table 1 antibiotics-13-00103-t001:** Examples of common specific isolation sources and distribution of isolation year within each category of isolation source.

Isolation Source	Animal	Environmental	Food	Human
Examples of common sources	Cattle, chicken, sheep, swine, canine, feline, horse, giraffe, gorilla; abscess, urine, feces, bile	Water, soil, environmental swab	Ground beef, raw turkey, raw chicken breast, retail milk, lettuce	Clinical samples: blood, feces
Number of strains isolated between 1947–2004	31	0	8	70
Number of strains isolated between 2005–2015	74	6	11	252
Number of strains isolated between 2016–2022	188	315	34	42
Number of strains with no known isolation year	3	6	1	408
Total number of strains	296	327	54	772

## Data Availability

The data presented in this study are openly available in NCBI GenBank. The isolate accession numbers are available in the Appendix A.

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
