# Peer review of "Associations between Isolation Source, Clonal Composition, and Antibiotic Resistance Genes in Escherichia coli Collected in Washington State, USA"

_antibiotics, 2024, doi:10.3390/antibiotics13010103_

Round 1

Reviewer 1 Report

Comments and Suggestions for Authors

I have carefully reviewed the manuscript titled "Association between Isolation Source, Clonal Composition, and Antibiotic Resistance Genes in Escherichia coli Collected in Washington State, USA" by Jewell et al. I appreciate the authors' efforts to investigate the patterns of antimicrobial resistance (AMR) across various sectors in Washington State using whole genome sequencing (WGS) of E. coli isolates. However, I have identified several areas that require attention before the manuscript can be considered for publication.

1.        The format of Line 103 and Table 1 is incorrect. You are advised to change the format to a three-line table.

2.        Line 39, Line 125, Line 133, Line 146, Line 165, Line 167, Line 169, Line 170, Line 172 , Many p values in the manuscript are not in italics, it is recommended to check and modify;

3.        The clarity of all figures in the article is low, so it is suggested to adjust them;

4.        The format of Line 195 and “blaCTX-M-15” is incorrect. It is recommended that they be consistent with the previous text.

5.        The reference quotation marks in "3. Discussion" and "4. Conclusions" are all bold, and it is suggested to adopt the same format as other parts of the paper.

In conclusion, the manuscript presents valuable insights into the association between isolation source, clonal composition, and antibiotic resistance genes in E. coli. However, addressing the above-mentioned concerns would significantly improve the quality and impact of the study. I recommend that the authors revise the manuscript to address these issues before considering it for publication.

Comments on the Quality of English Language

Minor editing of English language required.

Author Response

  1. The format of Line 103 and Table 1 is incorrect. You are advised to change the format to a three-line table.

As requested, the format of Table 1 has been changed to be a three-line table.

  1. Line 39, Line 125, Line 133, Line 146, Line 165, Line 167, Line 169, Line 170, Line 172 , Many p values in the manuscript are not in italics, it is recommended to check and modify;

Thank you for pointing this out. I believe all p-values are italicized now; I apologize for the oversight.

  1. The clarity of all figures in the article is low, so it is suggested to adjust them;

Thank you for this feedback. As suggested, the figures have been re-done to enhance clarity.

  1. The format of Line 195 and “blaCTX-M-15” is incorrect. It is recommended that they be consistent with the previous text.

Thank you for pointing this out. The formatting of “blaCTX-M-15” has been fixed.

  1. The reference quotation marks in "3. Discussion" and "4. Conclusions" are all bold, and it is suggested to adopt the same format as other parts of the paper.

All bracketed references throughout the paper are in bold now.

Reviewer 2 Report

Comments and Suggestions for Authors

In the manuscript titled "Association Between Isolation Source, Clonal Composition, and Antibiotic Resistance Genes in Escherichia coli Collected in Washington State, USA" authors studied the  important topic of antibiotic resistance, exploring the intricate relationship between sample sources and the presence of Antibiotic Resistance Genes (ARGs) in E. coli isolates from Washington State. The authors employed a robust methodology, tapping into public databases to gather whole genome sequences and conducting a thorough sequence analysis to uncover sequence types and resistance genes.

However, there are a few areas in the current manuscript that could benefit from some thoughtful refinement.

1. Section 2.3: It seems that the designation of isolates as Multidrug-Resistant (MDR) might be a little redundant since MDR isolates naturally fall within the resistant category. I'd recommend modifying the calculations to align with this understanding for a clearer presentation.

2. Discussion and Conclusion Sections: The discussion section is not prepared well. I'd suggest a complete rewrite, in clear comparisons with relevant previous studies. Discuss the analyses of each sample type and provide their implications. Additionally, the conclusion section appears to blend with the discussion, so it might be beneficial to carefully designate the two. Make the discussion rich and descriptive, while the conclusion should be crisp, summarizing the key takeaways of the study.

3. Section 4.5: The authors present a hypothesis from a One Health perspective, but there's a notable omission of environmental factors. It would be great to expand the scope beyond human and animal antibiotics, considering the potential impact of agricultural, aquacultural, and other practices. A more inclusive analysis, considering various factors contributing to selection pressure for Antimicrobial Resistance (AMR) emergence, will strengthen the paper.

Comments on the Quality of English Language

Satisfactory

Author Response

  1. Section 2.3: It seems that the designation of isolates as Multidrug-Resistant (MDR) might be a little redundant since MDR isolates naturally fall within the resistant category. I'd recommend modifying the calculations to align with this understanding for a clearer presentation.

Thank you for pointing this out. The categories of “resistant” and “multidrug resistant (MDR)” are meant to be mutually exclusive. The definitions of these categories have been clarified in lines 340-342 to address this confusion.

  1. Discussion and Conclusion Sections: The discussion section is not prepared well. I'd suggest a complete rewrite, in clear comparisons with relevant previous studies. Discuss the analyses of each sample type and provide their implications. Additionally, the conclusion section appears to blend with the discussion, so it might be beneficial to carefully designate the two. Make the discussion rich and descriptive, while the conclusion should be crisp, summarizing the key takeaways of the study.

In response to this feedback, we have re-structured and re-written parts of the Discussion and Conclusions. The Discussion is now richer with more comparisons to other studies and information about the implications of our findings, while the conclusion is shorter and focused on the main points of our study.

  1. Section 4.5: The authors present a hypothesis from a One Health perspective, but there's a notable omission of environmental factors. It would be great to expand the scope beyond human and animal antibiotics, considering the potential impact of agricultural, aquacultural, and other practices. A more inclusive analysis, considering various factors contributing to selection pressure for Antimicrobial Resistance (AMR) emergence, will strengthen the paper.

Thank you for this feedback. We agree that more data on environmental factors would strengthen this analysis, but unfortunately, the GenBank database contains a notable lack of both sequences and metadata from environmental sources, so we are unable to do further analysis on this topic. However, we have made note of this in our discussion as a limitation of the study (lines 225-229).

Reviewer 3 Report

Comments and Suggestions for Authors

antibiotic resistance in E. coli isolates from Washington State, USA.

I have no major comments for this study, which is useful and contributes to our knowledge regarding potential risk factors for antibiotic resistance.

Some points for changing are listed below.

-The authors should make very clear that the work is bioinformatics-based only and themselves did not contribute with own strains.

-Can the difference in relative frequency of STs between the four isolation source categories, be explained in greater depth, please?

-2.5. Can you please construct a heatmap with the frequencies of the 23 antibiotic profiles detected during the study?

-The Discussion is rather shallow and really does not do justice to this study. It must be extended and further aspects and implications of the findings should be highlighted.

Author Response

-The authors should make very clear that the work is bioinformatics-based only and themselves did not contribute with own strains.

Thank you for pointing this out. We have included a statement about this on lines 80-82.

-Can the difference in relative frequency of STs between the four isolation source categories, be explained in greater depth, please?

In response to this feedback, we added additional detail to this section (lines 116-128).

-2.5. Can you please construct a heatmap with the frequencies of the 23 antibiotic profiles detected during the study?

Thank you for this suggestion. A heatmap displaying this data is now included on page 7.

-The Discussion is rather shallow and really does not do justice to this study. It must be extended and further aspects and implications of the findings should be highlighted.

In response to this feedback, we have re-structured and re-written parts of the Discussion and Conclusions. The Discussion is now richer with more comparisons to other studies and information about the implications of our findings, while the conclusion is shorter and focused on the main points of our study.

Round 2

Reviewer 2 Report

Comments and Suggestions for Authors

I am satisfied with the revised version.

Comments on the Quality of English Language

Satisfactory